health and disease and epidemiology

infectious disease, externality, health aid, SIR model, Solow model

**Author for correspondence:**
Kamal Jnawali
e-mail: kamal.jnawali@oswego.edu

# Mitigating the externality of diseases of poverty through health aid

Kamal Jnawali[1], Michael G. Tyshenko[2] and Tamer Oraby[3]

[1]Department of Mathematics, State University of New York at Oswego, Oswego 13126-3599, NY, USA
[2]Risk Sciences International, Ottawa, Canada
[3]School of Mathematical and Statistical Sciences, University of Texas—Rio Grande Valley, Edinburg, TX, USA

KJ, 0000-0001-6538-2001; TO, 0000-0002-8176-1324

Externality exists in healthcare when an individual benefits from others being healthy as it reduces the probability of getting sick from illness. Healthy workers are considered to be the more productive labourers leading to a country's positive economic growth over time. Several research studies have modelled disease transmission and its economic impact on a single country in isolation. We developed a two-country disease-economy model that explores disease transmission and cross-border infection of disease for its impacts. The model includes aspects of a worsening and rapid transmission of disease juxtaposed by positive impacts to the economy from tourism. We found that high friction affects the gross domestic product (GDP) of the lower-income country more than the higher-income country. Health aid from one country to another can substantially help grow the GDP of both countries due to the positive externality of disease reduction. Disease has less impact to both economies if the relative cost of treatment over an alternative (e.g. vaccination) is lower than the baseline value. Providing medical supplies to another country, adopting moderate friction between the countries, and finding treatments with lower costs result in the best scenario to preserve the GDP of both countries.

## 1. Introduction

Diseases of poverty include infectious diseases such as tuberculosis (TB), malaria and HIV/AIDS, as well as diseases stemming from malnutrition. Poverty is a main social determinant of health in developing countries and diseases of poverty are often correlated with malnutrition [1]. It was estimated that diseases of poverty account for almost half of the

burden of diseases in countries with the highest poverty rates. Despite being manageable with existing treatments and often preventable, diseases of poverty exact an enormous amount of mortality and morbidity toll on the populations in low-income countries [2].

Labour participation and productivity of a country are determined by the interaction of the population's exposure to diseases and how it impacts human capital accumulation over time. Previous studies have shown that low-income countries, when compared with high-income countries, have less income to spend on disease risk management efforts to control disease transmission and reduce incidence. Infectious diseases affect the ability of individuals to work and reduces the productivity of the country's overall human capital [3].

High-income countries possess infrastructure, training and the ability to build disease surveillance capacity for a new disease if it is lacking. Epidemiology with contact and case tracing can inform appropriate resource allocation by using the evidence base from health-related incidents in a population. Surveillance and epidemiology allow for rapid response to disease occurrence, recurrence, the number of incidents and the rate at which the disease transmits within the population. Epidemiological evidence helps to inform disease risk management policies used by risk managers and to evaluate the efficacy of programmes and intervention policies [4].

A poverty trap refers to conditions that may trap individuals, groups and whole economies in intractable poverty. Also, poverty is considered one of the greatest risk factors for acquiring endemic infectious diseases. The resulting chronic morbidities and co-morbidities act to reduce human capital and hamper efforts to escape diseases of poverty, and results in an ongoing poverty trap. There are various research studies addressing the insidious cycle between prevalence of diseases and the occurrence of poverty traps [5,6]. Economic and epidemiological conditions are indicators of the overall health and socio-economic growth potential of the population, so the poverty trap can be weakened by action that enhances the health status of the population [5]. Disease and poverty have a constant, nonlinear relationship and there is a significant link between the infectious diseases of poverty and poverty-related conditions like the lack of sanitation and nutrition that make people more vulnerable to infections and diseases in general. A recent study showed the relationship between death from non-communicable disease (NCD) and poverty [7]. Health is an important element of economic prosperity and we only need to look at the perpetual poverty of African countries and the prevalence of endemic diseases to understand the negative impacts. Diseases such as malaria cause an externality affecting human capital and it also may be a factor that contributes to disease persistence and economic decline [8,9].

The term externality is defined in a number of ways. Externality is a result when an agent produces or consumes a good or service that causes a cost or benefit on other parties who are not involved in this transaction [10]. In terms of disease spread, it could be identified through people's risk of infection due to other people's behaviours [11]. An externality can be both positive and negative, as in the form of a benefit or cost. An economic approach of unpaid environmental impacts on production and consumption that influence the cost and utility of consumers over market functioning comes under environmental externalities. Whereas, disease externalities begin from an individual's action while adopting prevention and treatment of infectious disease that finally affects others in terms of their health and choices. At the population level, externalities can be reduced by implementing interventions, such as taxes, subsidies and new measures.

Many researchers have studied infectious disease using modelling and economics, but these are often done in isolation [12–14]. A few research studies have shown the impacts from infectious diseases on a country's economy but remain focused largely on the dynamics within the country, in isolation from other populations [15–18]. However, there are conditions under which a country's economy can be severely affected by health choices that are adopted by individuals of another country due to externalities. For example, the recent COVID-19 pandemic has highlighted how individual country choices to limit the movement of goods and people both within and between countries has affected, in part, the timing, introduction, and severity of the outbreaks from travel-related transmissions of SARS-CoV-2 [19].

It is well known that poor people in low-income countries suffer from higher rates of illness, particularly infectious diseases and malnutrition due to several factors including little food, unclean water, low levels of sanitation and shelter, lack of infrastructure to deal with high exposure to infectious agents and lack of appropriate medical care [20]. Research has confirmed that addressing determinants of health can yield significant sustainable returns for improving population health [21]. The ability of the healthcare system to provide effective services can be strengthened by various types of health aid such as training (increasing the availability of adequate skilled healthcare workers, e.g. nurses, doctors), infrastructure (essential equipment, supplies, health facilities) and medicines (such as vaccines, antivirals and drugs) to meet the needs of the population they assist [20]. Our analysis

focuses on the provision of health aid given in the form of vaccines as an example. This health aid addresses and manages longer-term transmissible disease impacts in the context of our model where externality is caused by the movement of infected people between two countries.

We developed a disease transmission model within and between two hypothetical, neighbouring countries and show its impact on the gross domestic product (GDP). Under our simulations, a non-specific infection is shown entering the country where it becomes endemic circulating within the population repeatedly. For example, a non-specific infection could be a disease of poverty such as tuberculosis (TB), which has been circulating in India for at least a few hundred years [22]. We explore how mobility between countries and communities can affect the respective economies through disease transmission and how health aids can help to break the effects of the externality. If we consider that a number of adults within a country become infected from a disease then it can drastically affect the GDP of that country. If one country gets the disease, then another country may restrict travel completely as a way of preventing the importation of cases. Prolonged travel restrictions between countries can exert a negative effect decreasing the GDP of both countries. The optimal situation may be obtained by providing adequate health aid to another country, such as building surveillance capacity, medical supplies, antivirals and vaccines, while attempting to mitigate the transmission of the disease or prevent the disease transmission and protect its GDP.

# 2. Model and methods

## 2.1. Modelling disease and economic growth

Our model introduces a two-country disease-economy coupled system of differential equations. The coupled system is used to study an infectious disease starting in country 1 and its epidemiological and economic repercussions on country 2, and then back on itself due to cross-border infection. In particular, we consider that the two exclusively interacting countries are initially at two different socio-economic levels and of different population sizes. Disease spreads internally in country 1 and is transmitted to country 2 and then back to country 1 due to the mutual interaction of citizens (e.g. visitation such as tourism) modelled using the gravity model of trade [23]. At the same time, visitation adds to the economy growth of the destination country [23].

We consider both non-mandatory vaccination and treatment interventions for the disease to be available and in turn their degree of use affects the GDP. Also, some infected individuals are not considered as part of the working labour force due to extended illness, especially at high GDP levels. Treatment can decrease the length of the illness time period. People voluntarily choose to be vaccinated based on their perception of risk of the disease, and it is assumed to be proportional to disease prevalence and outcomes [24].

Important for understanding the interaction of disease and its cross-border infection, we study the effect of health aid on the disease and economic progress in both countries if health aid from country 2 into country 1 is implemented. Health aid in our scenarios is represented by the parameter $A$ defined as the number of vaccines subsidized per year by country 2 for country 1, during the existence of the disease in country 1.

We use a pair of disease models based on a susceptible-infected-recovered (SIR) model (see appendix A)

$$\frac{\mathrm{d}}{\mathrm{d}t}x_i = \mu_i(1 - x_i) - p_i x_i - x_i(\lambda_{i,i}y_i + \lambda_{i,j}y_j) \tag{2.1}$$

and

$$\frac{\mathrm{d}}{\mathrm{d}t}y_i = x_i(\lambda_{i,i}y_i + \lambda_{i,j}y_j) - \gamma_i(1 + \Lambda_i)y_i - \mu_i y_i \tag{2.2}$$

for $i = 1, 2$ and $j = 2, 1$; The disease model is coupled with a pair of economic growth models (Solow model) of the *per capita* GDP ($1000) given by the equation (see appendix A)

$$\frac{\mathrm{d}}{\mathrm{d}t}k_i = \sigma_i k_i^{\alpha_i} l_i^{1-\alpha_i} - (\delta_i + \mu_i - \nu_i)k_i - c_V(t)((p_i x_i) + \tau*(\gamma_i \Lambda_i y_i)) + \sigma_{ij}T(k_i, N_i; k_j, N_j)\frac{N_j}{N_i} \tag{2.3}$$

for $i = 1, 2$ and $j = 2, 1$. The two systems start initially at $y_1(0) > 0$, $y_2(0) = 0$, $k_1(0)$, and $k_2(0)$ are at different levels. The variables $x_i$, $y_i$ and $N_i$ are the density of susceptible, prevalence and population size of country $i$,

respectively. The variables $k_i$ and $l_i$ are the *per capita* GDP ($1000) and fraction of labour to population size of country $i$, respectively. Birth and death rates are given as $\mu_i$ and $v_i$, see appendix B. Recovery rate and transmission rates are given as $\gamma_i$ and $\lambda_{i,j}$. Treatment relatively increases the recovery rate by $\Lambda_i$. All of the parameters $\mu_i$, $v_i$, $\gamma_i$, $\lambda_{i,j}$, $\Lambda_i$ functionally depend on $k_i$, see appendix B. The probability of vaccination is $p_i = \theta_p\, y_i$ as a rule-of-thumb for the decision to get vaccinated. The rate of capital accumulation due to production and visitation are given as $\sigma_i$ and $\sigma_{i,j}$. The parameter $\alpha_i$ is the constant of elasticity in the Cobb–Douglas form of production with constant to returns scale and $\delta_i$ is the depreciation rate. $T(k_i, N_i; k_j, N_j)$ is the gravity function of country $j$ into the country $i$ based on *per capita* GDP, population sizes and friction.

The model also includes the cost of vaccination $c_V(t)$ and treatment $c_T(t)$. The cost of treatment is a $\tau$ multiple of the cost of vaccination $c_T(t) = \tau * c_V(t)$. Since the disease evolution and development process are modelled in a time frame of decades, we use an inflation factor for the cost of vaccination with a rate $f$. Finally, $\ell(k)$ is the proportion of infected individuals who are available for work, probably as a result of the difficulty to have a paid leave of absence due to low GDP. A full account of derivation of equations (2.1)–(2.3) and their parameters are given in appendices A and B.

We use birth and death rates that are dependent on the socio-economic status of the countries based on estimates similar to those made previously [25]. Disease transmission and recovery rates are dependent on the socio-economic status of the two countries through sanitation and nutrition, respectively, that were also previously estimated [25], (see also appendix B for adjustments that we made to these rates). The basic transmission rate $\beta_0$ is the maximum transmission rate achievable at zero (practically, very low) GDP. It is a multiplication of the basic contact rate $\lambda_0$ and the probability of transmission $\rho_0$ upon contact when there is no sanitation at all. The basic recovery rate $\gamma_0$ is the maximum recovery rate achievable for an infinite (practically, very large) GDP due to high nutrition levels, see appendix B. The basic treatment effect $\Lambda_0$ is the maximum physiologically possible relative increase in the recovery rate achievable for an infinite (practically, very large) GDP due to better medical care infrastructure, see appendix B.

We also examine the effect of health aid on the economy of both countries by comparing the cost of vaccination and treatment against subsidizing vaccination in country 1. This could be done by introducing a number of compulsory vaccines subsidized per year, $A$, by country 2 for country 1 whenever the disease exists. Thus, the equation describing the rate of change in $x_1$ becomes

$$\frac{\mathrm{d}}{\mathrm{d}t}x_1 = \mu_1(1 - x_1) - p_1 x_1 - I_{(y_1>0)}A/N_1 - x_1(\lambda_{1,1}y_1 + \lambda_{1,2}y_2), \tag{2.4}$$

and the equation describing the rate of change in $k_2$ becomes

$$\frac{\mathrm{d}}{\mathrm{d}t}k_2 = \sigma_2 k_2^{\alpha_2} l_2^{1-\alpha_2} - (\delta_2 + \mu_2 - v_2)k_2 - c_V(t)*((p_2 x_2 + I_{(y_1>0)}A/N_2) + \tau*(\gamma_2 \Lambda_2 y_2)). \tag{2.5}$$

## 2.2. Model calibration

We used available tuberculosis (TB) data (of all types) from Mexico and the USA, from 1990 to 2010, to calibrate the model parameters [26]. Here, we use Mexico and the USA as examples for calibrating the model with $N_1(0) = 87\,065\,000$ and $N_2(0) = 248\,709\,873$, respectively. We fitted the time series of infections and i GDP ($1000) to the model's simulation. (While this is in no way an actual estimation of model parameters, we use the best-fit parameters values as a starting point for the analyses.) We used weighted sums of squares of errors as the objective function in the minimization problem. We implemented the minimization process using the genetic algorithm function available in Matlab. We used the calibration process to find initial values of *per capita* GDP $k_1(0) = 3.112$ per 100 000, $k_2(0) = 23.889$ per 100 000, and prevalence $I_1(0) = 16.78$ per 100 000, and $I_2(0) = 11.325$ per 100 000. The list of parameters that were found through calibration and the values are given in table 1. Other parameters discussed in appendix B were fitted to actual data from external sources like Gapminder.com. See appendix C for results of model calibration and parameter values.

## 2.3. Model analysis

We let the *per capita* GDP $k_i(t)$ and prevalence $y_i(t)$ be represented as functions in a vector parameter $\theta$, and denote them by $k_i(t; \theta)$ and $y_i(t; \theta)$, respectively. As a measure in the change of the course of the epidemic and economy's growth, we use the area under the curve over a length of time $T$, ([27],

**Table 1.** Model's parameters, their description and base values.

| parameter | description | base values | range | sources |
|---|---|---|---|---|
| $\beta_0$ | basic transmission rate | 0.458 | (0.3, 0.6) | calibrated |
| $\tau$ | relative cost of treatment to cost of vaccination | 8191.88 | (1000, 10 000) | calibrated |
| $\gamma_0$ | base recovery rate | 0.396 | (0.2, 0.5) | calibrated |
| $\Lambda_0$ | maximum percentage increase in recovery rate due to treatment | 0.346 | (0.2, 0.5) | calibrated |
| $\phi$ | rate of decrease in percentage increase in recovery rate due to treatment | 0.04 | (0.01, 0.1) | calibrated |
| $\theta_p$ | constant of proportionality in probability to get voluntarily vaccinated | 0.01 | (0.001, 1) | calibrated |
| $\hat{\sigma}_i$ | rate of capital accumulation due to production in country $i$ | {0.224, 0.342} | (0.1, 0.5) | calibrated |
| $\hat{\sigma}_{ij}$ | rate of capital accumulation due to visiting country $i$ by people from country $j$ | {0.046, 0.021} | (0.01, 0.05) | calibrated |
| $\alpha_i$ | Capital's and labour's shares (elasticity) of output in the Cobb–Douglas formula for country $i$ | {0.491, 0.492} | (0.45, 0.55) | calibrated |
| $\ell$ | fraction of labour who is working while infected | 0 | (0, 1) | assumed |
| $\delta_i$ | depreciation rate of wealth in country $i$ | {0.035, 0.021} | (0.01, 0.05) | calibrated |
| $\mu_i$ | birth rate in country $i$ | a function in GDP $k$ | — | see appendix B |
| $\nu_i$ | death rate in country $i$ | a function in GDP $k$ | — | see appendix B |
| $F_{ij}$ | the friction people from country $j$ face when they move to country $i$ | $F_{1,2} = 0.037$ and $F_{2,1} = 0.018$ | (0.01, 0.05) | calibrated |
| $c$ | gravity law-constant | 0.764 | (0.5, 1) | calibrated, see also [31] |
| $\xi_0$ | gravity law—elasticity of source's population size | −0.116 | (−0.15, −0.05) | calibrated, see also [31] |
| $\xi_1$ | gravity law—elasticity of relative *per capita* GDP | 0.154 | (0.1, 0.2) | calibrated, see also [31] |
| $\xi_2$ | gravity law—elasticity of destination's population size | 0.686 | (0.5, 1) | calibrated, see also [31] |
| $\xi_3$ | gravity law—elasticity of friction | 3.601 | (1, 5) | calibrated, see also [31] |
| $f$ | annualized inflation rate | 0.043 | (0.01, 0.1) | calibrated |

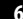

**Figure 1.** Simulation of externality and cross-border infection of diseases for different degrees of contagiousness when both countries are initially equally sized. Simulations of GDP (in $1000) and disease prevalence for country 1 (a,c,e) and country 2 (b,d,f) when an epidemic starts in country 1 with 10 initial cases, $\tau = 1000$ and $N_1(0) = N_2(0) = 10^7$. The three rows of figures are simulations with (a,b) $k_1(0) = 9$, and $k_2(0) = 55$; (c,d) $k_1(0) = 9$, $k_2(0) = 9$, and $\sigma_2 = \sigma_1 = 0.224$; and (e,f) $k_1(0) = 55$, $k_2(0) = 55$, and $\sigma_1 = \sigma_2 = 0.432$. Values of the rest of the parameters are given in appendix C and table 1.

p. 340). In particular, we use two measures defined on the curve of simulation $X(t; \theta)$; e.g. *per capita* GDP $k_i(t; \theta)$ up to time $T$. The first is the area under the curve of $X$

$$\text{AUC}(X|\theta) = \int_0^T X(t; \theta)\, \mathrm{d}t,$$

for the vector of parameters $\theta$. We also define the ratio of the areas under the curves at two vector values of the parameters $\theta_1$ and $\theta_2$ to be

$$\text{RAUC}(X|\theta_1, \theta_2) = \frac{\text{AUC}(X|\theta_2)}{\text{AUC}(X|\theta_1)}.$$

We use a time window of a century, $T = 100$, as the base for our investigation.

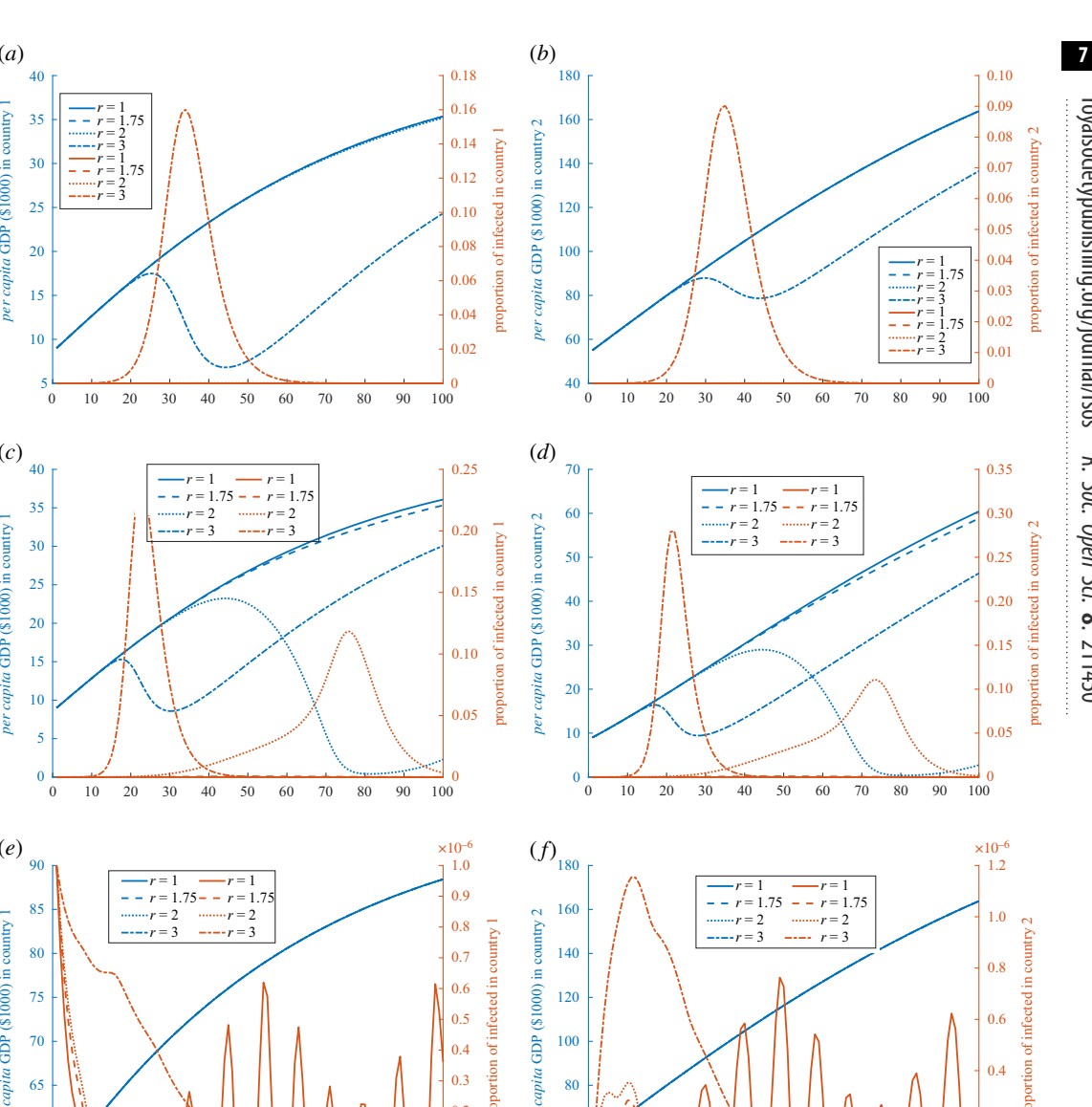

**Figure 2.** Simulation of externality and cross-border infection of diseases for different degrees of contagiousness when both countries are initially not equally sized. Simulations of GDP (in \$1000) and disease prevalence for country 1 (a,c,e) and country 2 (b,d,f) when an epidemic starts in country 1 with 10 initial cases, $\tau = 1000$, $N_1(0) = 10^7$ and $N_2(0) = 10^8$. The three rows of figures are simulations with (a,b) $k_1(0) = 9$, and $k_2(0) = 55$; (c,d) $k_1(0) = 9$, $k_2(0) = 9$, and $\sigma_2 = \sigma_1 = 0.224$; and (e,f) $k_1(0) = 55$, $k_2(0) = 55$, and $\sigma_1 = \sigma_2 = 0.342$. Values of the rest of the parameters are given in appendix C and table 1.

Integrals are numerically calculated using the trapezoid rule. The integrands are solutions of the ordinary differential equation (ODE) system, which are also solved numerically using the Runge–Kutta method. All numerical procedures are implemented using Matlab.

# 3. Results

## 3.1. Model simulation

We ran simulations of the model to investigate the effect of some important parameters on the externality and cross-border infection on both countries. We let the basic transmission rate be an $r$ multiple of the calibrated value in table 1; that is, $\beta_0 = 0.46r$ for $r \geq 1$. In the following simulations, we examined the effects of the initial economy status, population size, friction and cost of treatment on the disease

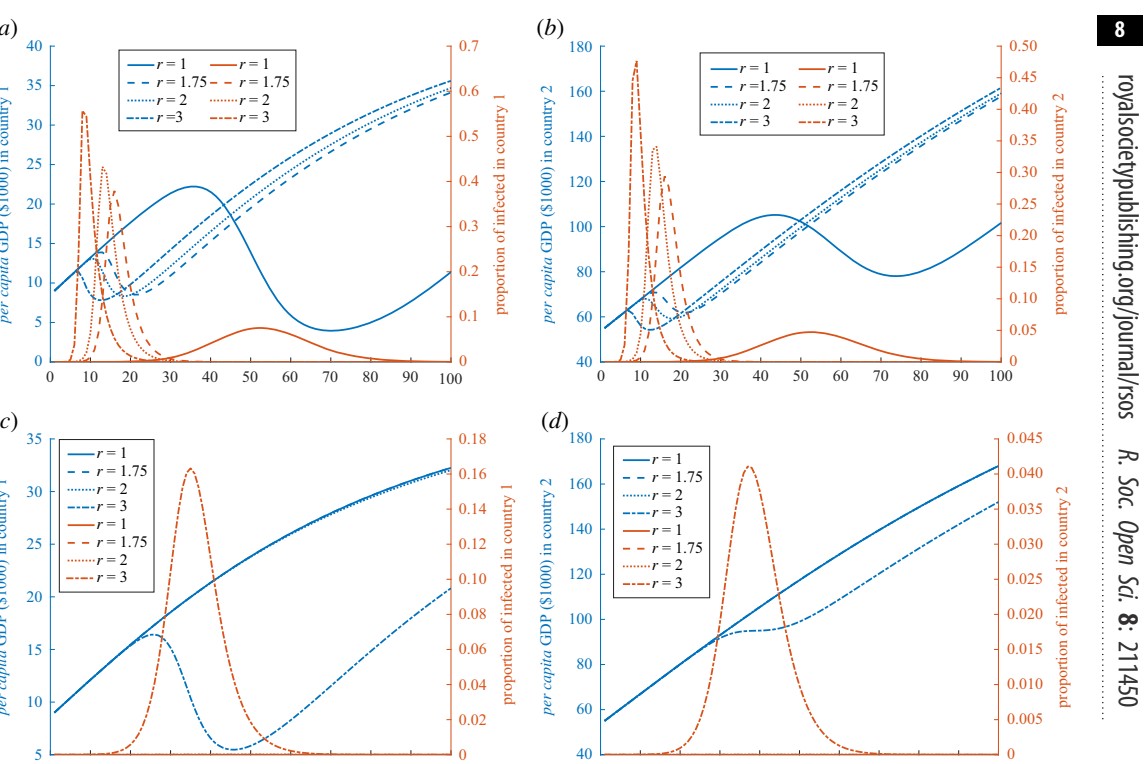

**Figure 3.** Simulation of externality and cross-border infection of diseases when friction between countries are the same. Simulations of GDP (in $1000) and disease prevalence for country 1 (*a,c*) and country 2 (*b,d*) when an epidemic starts in country 1 with 10 initial cases, $\tau = 1000$, $k_1(0) = 9$ and $k_2(0) = 55$, and $N_1(0) = N_2(0) = 10^7$. Fraction between countries are (*a,b*) $F_{12} = F_{21} = 0.0178$, and (*c,d*) $F_{12} = F_{21} = 0.0374$. Values of the rest of the parameters are given in appendix C and table 1.

spread within and between the two countries and the induced change in economy. We then examined the counter-effect of health aid on the disease and economy.

Throughout the simulations, the disease spreads differently in both countries, but the qualitative features of the epidemics stay the same. In many cases, the economy receives a shock that it doesn't recover from over a very long time, so much so, that the economy seems entrapped in poverty. In figure 1, we introduce a baseline scenario for countries with equal population sizes, while the initial GDP and rate of capital accumulation are varied to capture the trend of disease prevalence and the GDP for different degrees of contagiousness $r \geq 1$. When one of the two countries is rich or developed, at smaller values of $r$, it takes the prevalence a longer time to reach its peak, and as a result the epidemic lasts longer for both countries, figure 1*a,b*. With a moderate magnitude of epidemics, the length of the epidemics has profound effect on economy. The reason is that a fast growing epidemic triggers larger vaccination uptake (under the assumption that a vaccine is readily available) and so the outbreak subsides faster and results in a resumption in economic development. Slowly increasing epidemics, moderated by the waning nutrition and sanitation as economy drops, may not be perceived to be as risky and fail to trigger vaccination uptake in the population, which can affect the economy immensely in a negative feedback loop. In this case, at the beginning, moving from less developed country 1 to the more developed country 2 is greater for better economic opportunities, an effect magnified by low friction. But when both countries are initially at the same economic status, the effect on the economy is less drastic but still significant if both were initially poor, figure 1*c,d*. If both are rich countries they would be able to absorb the effect of the externality and cross-border infection, figure 1*e,f*. This seems to be true due to their preparedness and infrastructure that can deal with the outbreak magnitude, through sanitation, nutrition, and due to less movement between the countries.

The effects of contagiousness changes as country 2 becomes 10 times larger than country 1, at least when one of the two countries is poor. Only high levels of infectiousness can hurt the economic development in the poor/rich and poor/poor situations, figure 2*a,b*. Disease spreads quickly and halts quickly as well in poor/rich countries. But, disease hits the economy harder when both countries are

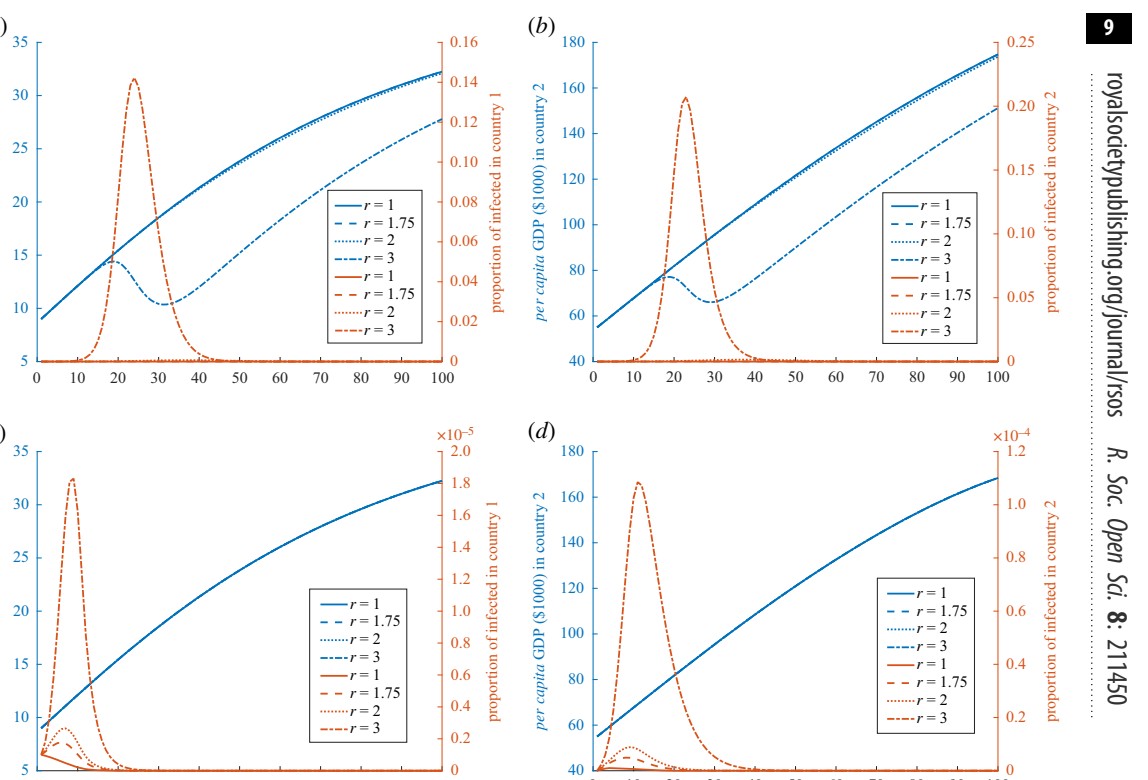

**Figure 4.** Simulation of externality and cross-border infection of diseases when aid is given from country 2 to country 1. Simulations of GDP (in $1000) and disease prevalence for country 1 (*a*,*c*) and country 2 (*b*,*d*) when an epidemic starts in country 1 with 10 initial cases, $\tau = 1000$, $k_1(0) = 9$ and $k_2(0) = 55$, and $N_1(0) = N_2(0) = 10^7$. Aid from country 2 to country 1 is (*a*,*b*) $A = 10^5$, and (*c*,*d*) $A = 10^6$. Values of the rest of the parameters are given in appendix C and table 1.

poor and they could become even poorer if the disease takes a long time to cease, especially when it is less infectious and so slowly spreading, figure 2*c*,*d*. If both countries are initially rich, then their economy will not be affected as disease will find resistance to spread between people, figure 2*e*,*f*. A greater population size in country 2 produces multiple waves of the epidemic in country 2 and spills back to country 1 as well for moderate values of *r* (*r* = 1.75 or 2), figure 2*e*,*f*. Since the countries are richer and have higher capital accumulation, the GDP remains unaffected by the degree of contagiousness, *r*.

Low friction between countries can cause damage to both economies worse than when the disease is not very infectious, figure 3*a*,*b*. Along with the effect of higher friction between the two countries, figure 3*c*,*d* shows the disease externality and cross-border infection effects of diseases influences on the economy in different ways. It looks as if it can affect the poor country to a greater degree, for high infectiousness, if there is high friction rather than with low friction, figure 3*a*,*c*.

Health aid, from country 2 to country 1 can substantially save the economy of both countries, figure 4*a*,*b* in contrast to figure 1*a*,*b*. Enhanced health aid from $A = 10^5$ to $A = 10^6$ from country 2 to country 1, curbs the epidemic in country 1 by approximately 99.7% and saves them both from externality, figure 4*c*,*d*. As a result, the economies of both countries grow steadily without disruption.

A lower cost of treatment as an intervention ($\tau = 100$) shows that disease has less effect on economic growth for both countries, figure 5*a*,*b*. However, in case of changing $\tau = 10\,000$, the GDP of both countries almost collapse and take roughly 60 years to regain their initial GDPs, figure 5*c*,*d*. This can be the case of severe transient poverty that both countries would face due to an extreme spread of infection even for diseases with small transmission rates.

To understand the mutual effect of frictions, we simulated the model with $\delta_i = 0$ for $i = 1$, 2, and calculated the relative areas under the curve RAUC($k_i | (F_{12}, F_{12}, r = r^*)$, $(F_{12}, F_{12}, r = 1)$) for $i = 1$, 2 and $r^* = 1.75$, 2, 3. It seems that there are pairs of friction where the more contagious the disease is the worse is the impact of the disease on the economy (the yellow coloured region in figure 6*a*–*f*). There are also some regions where the less contagious becomes worse (the dark blue coloured region in figure 6*a*–*f*).

That might be also interpreted differently as that a decrease in effective contact rate, probably by social distancing, might be a recipe for saving the economy. But that would not be a reasonable

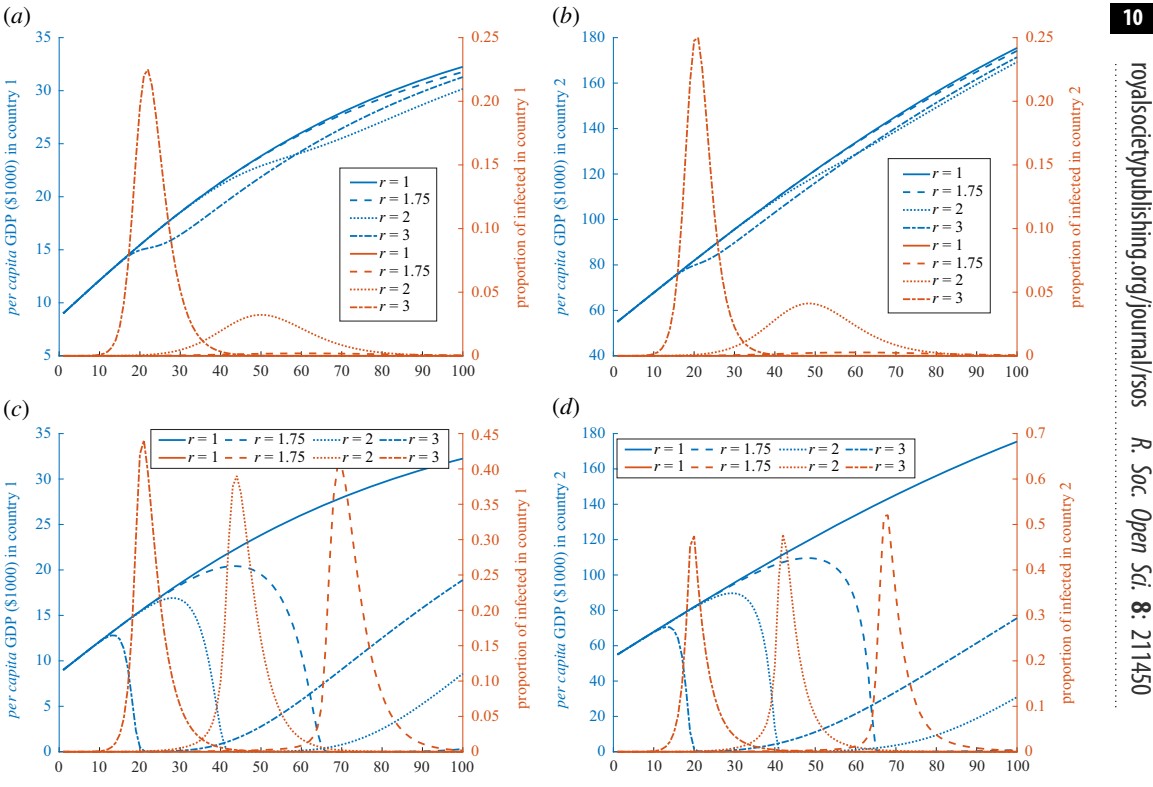

**Figure 5.** Simulation of externality and cross-border infection of diseases at different costs of treatment. Simulations of GDP (in $1000) and disease prevalence for country 1 (a,c) and country 2 (b,d) when an epidemic starts in country 1 with 10 initial cases, $k_1(0) = 9$ and $k_2(0) = 55$, and $N_1(0) = N_2(0) = 10^7$. Relative cost of treatment is (a,b) $\tau = 100$, and (c,d) $\tau = 10^4$. Values of the rest of the parameters are given in appendix C and table 1.

solution due to the long time horizon of the disease endemic. We, however, calculated the relative areas under the curve $\mathrm{RAUC}(k_i \,|\, (F_{12}, F_{12} \,|\, A = 0), (F_{12}, F_{12}, A = A^*))$ for $i = 1, 2$ and $A^* = 10^5, 10^6$ when $r = 3$, figure 7a–d. The larger the aid, in which case 10% of the first country is regularly vaccinated, will lead to similar results to decreasing contact by one-third, compare figures 6e,f and 7c,d.

The level of health aid and the timing have effects on the two countries' economies. We examined the effect of various times of starting the health aid programmes in the form of compulsory vaccination of people in country 1. Long delays in sending health aid (vaccines) or not sending aid at all result in loops that are due to the feedback cycle between disease and economy (figure 8a,b). A swift significant response is required to mitigate the economical loss of both countries in which case GDP and prevalence follow a regular inverted U-shaped function (figure 8a–d). A slightly late response, however, can still break through the loop, but to do so requires larger levels of aid (figure 8a–d).

## 4. Discussion

An externality begins when an individual is involved in an activity that affects the welfare of others who neither pay nor receive compensation for that activity. The existence of the externality is considered one of the crucial factors in healthcare that influences disease patterns. An individual benefits from others remaining healthy because it reduces the likelihood of disease transmission if the illness is contagious. The presence of externality needs authorities to implement an intervention, such as mass vaccination. A vaccinated person is less likely to be a disease carrier which prevents other individuals from becoming ill [28]. People may choose whether to vaccinate or not under the cost–benefit analysis that results in the externality within the healthcare system. The concept of externality is highly prevalent in economics; however, it is not widely applied to the healthcare sector.

We developed a two-country disease-economy model to analyse the cross-country disease propagation and its long-term effect on GDP. As a baseline scenario, we considered country 1 as a

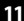

**Figure 6.** Parameter planes for fractions $F_{12}$ and $F_{21}$ for different degrees of contagiousness. Parameter planes to show the relative area under the curve of GDP (in \$1000) when $r = 1$ to $r = 1.75$ in (a,b), $r = 2$ in (c,d), and $r = 3$ in (e,f). Relative area under the curve of GDP (in \$1000) for country 1 (a,c,e) and for country 2 (b,d,f). In all of the calculations, epidemic starts in country 1 with 10 initial cases, $k_1(0) = 9$ and $k_2(0) = 55$, and $N_1(0) = N_2(0) = 10^7$. Relative cost of treatment is $\tau = 1000$. The depreciation rates are $\delta_1 = \delta_2 = 0$. Values of the rest of the parameters are given in appendix C and table 1.

poor country, where disease initiates and transmits to country 2 that is economically rich. In this scenario, increasing the degree of contagiousness increases the number of infections that, in turn, contributes to sharply declining GDPs for both countries. Countries do not regain their former GDP levels for 100 years. The countries are in a transient poverty state induced by chronic disease conditions, but they reach their supposed level in a very long term, data are not shown. The GDP of country 2 deteriorates due to externality since the infection that begin in country 1, and the GDP of country 1 worsens due to the steady cross-border infection effect of the disease, but still in a transient poverty state. More importantly, note that in figure 6a,c,e the larger the friction of movement from country 2 to country 1, the more ameliorated the effect of the disease on the economy of country 1 is, due to the cross-border infection effect. However, there is also lower friction that is also relatively less harmful to the economy, but not necessarily in magnitude. There might be situations where externality and cross-border infection have positive effects on eradicating the disease through modulated movements between the two countries.

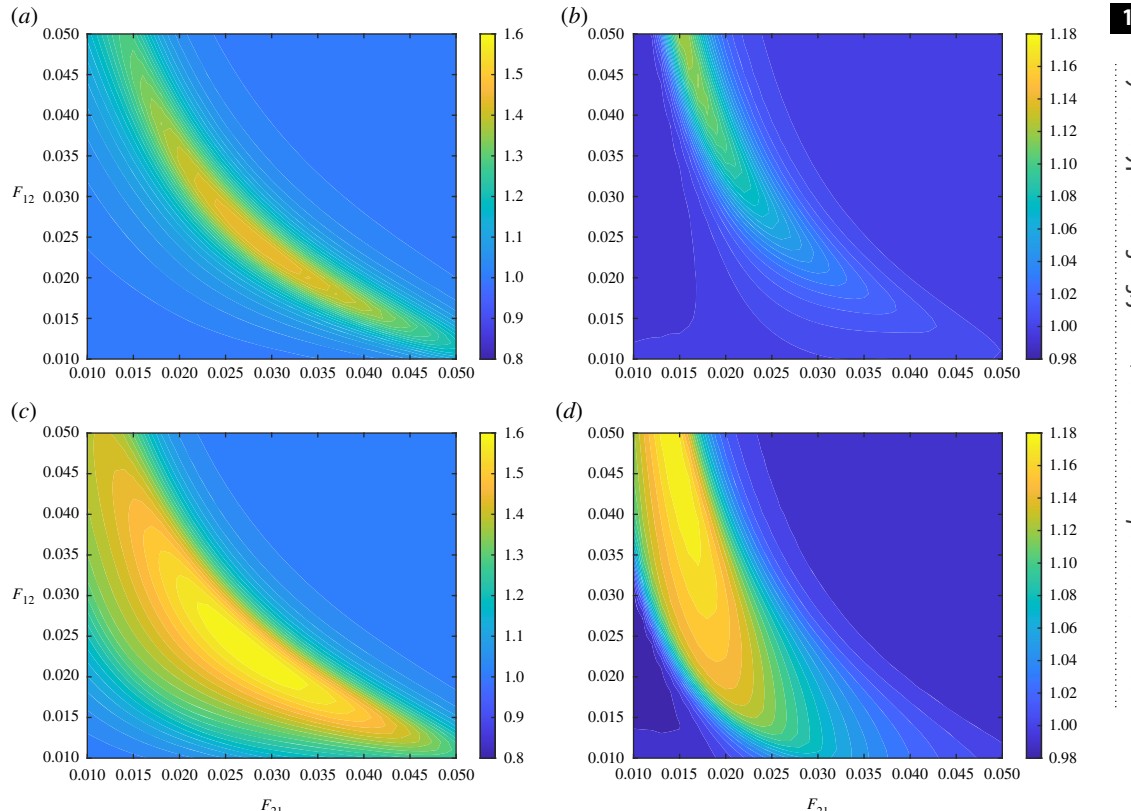

**Figure 7.** Parameter planes for fractions $F_{12}$ and $F_{21}$ for different levels of aid. Parameter planes to show the relative area under the curve of GDP (in \$1000) when $r = 3$ for $A = 10^5$ relative to $A = 0$ in ($a$,$b$), and $A = 10^6$ relative to $A = 0$ in ($c$,$d$). Relative area under the curve of GDP (in \$1000) for country 1 ($a$,$c$) and for country 2 ($b$,$d$). In all of the calculations, epidemic starts in country 1 with 10 initial cases, $k_1(0) = 9$ and $k_2(0) = 55$, and $N_1(0) = N_2(0) = 10^7$. Relative cost of treatment is $\tau = 1000$. The depreciation rates are $\delta_1 = \delta_2 = 0$. Values of the rest of the parameters are given in appendix C and table 1.

We allowed each of the parameters to vary individually to observe the actual variation and impact of that parameter to the model prediction. We explored the result of our simulations by creating several scenarios and economic conditions for two countries, such as both countries are poor, both countries are rich, and the countries have differing economic status (one poor and one rich). Equal and reduced friction causes the infection to grow higher, for even a small value of $r$, for both countries, as the huge flow of population across the countries spreads the disease faster. On the other hand, we observed that curtailing or lowering the friction establishes a 'give and take' situation for both countries since the visitation to another country boosts its economy. Providing swift and substantial health aid, such as a supply of vaccine from country 2 to country 1 completely prevents the disease from appearing in the country that receives the vaccine. The rate of transmission remains low, which allows the country to retain its stable prosperity in GDP throughout the simulation time for both countries. We consider vaccination as a form of health aid, but it could be one or a collection of measures of permanent health protection that have a synergistic effect.

The lower the value of relative price of treatment to vaccination $\tau$, the better the growth of the countries' GDP especially for low disease transmission rates. For higher $\tau$ values, the infection spreads with a significantly higher peak number of infections with a large impact on the GDP. If the cost of treatment is low it allows people to be able to treat themselves and recover faster where they stay relatively healthy, even though many people get infected with a higher transmission rate of the disease. Those who recover quickly minimize the loss of human capital which helps to protect their GDP from depreciating.

We have considered many variations on which we analysed the model from baseline scenario by altering various parameter values, one at a time. The optimal strategies for both countries to adopt may include, but are not limited to: reducing the friction through implementing soft immigration rules between the countries; promoting nutrition and hygiene practices in different countries, helping to improve sanitation infrastructure; providing medical support such as vaccines and vaccination programme development from a richer country to a poorer country; and finding lower costs of treatments over the cost of vaccinations.

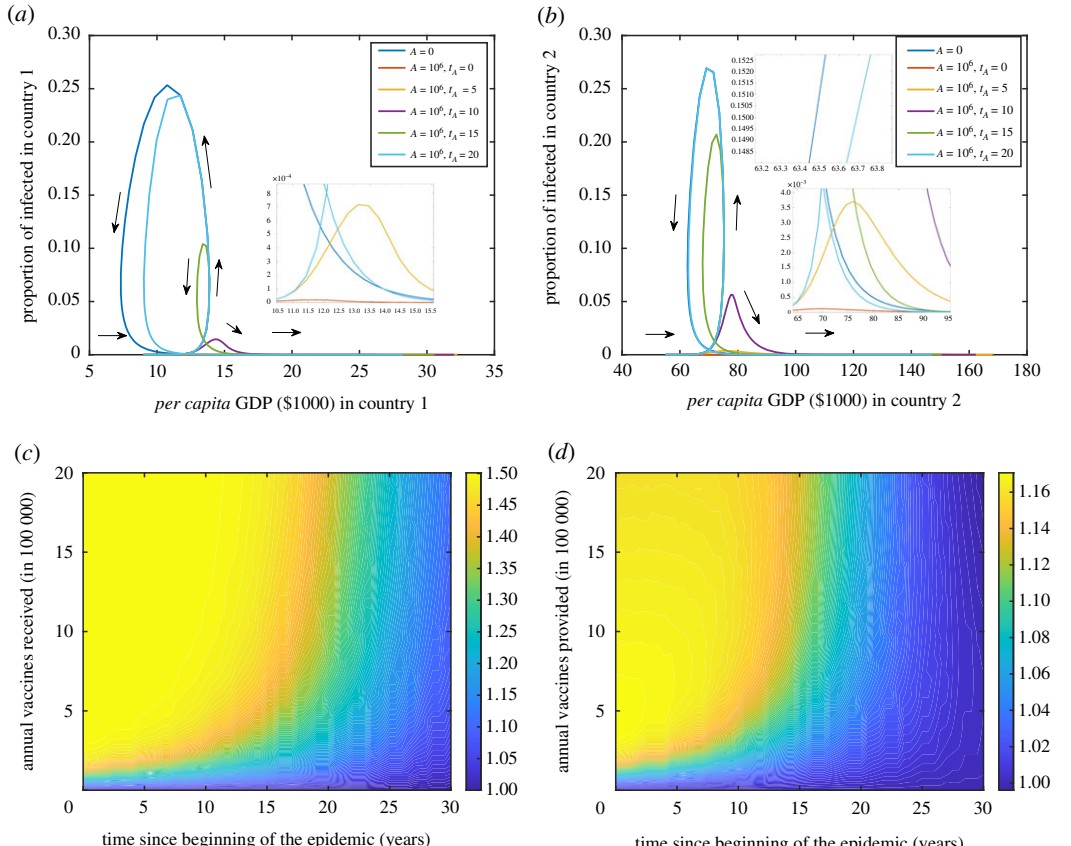

**Figure 8.** Health aid levels and timing. Phase diagram for *per capita* GDP ($1000) versus prevalence for country 1 in (*a*) and country 2 in (*b*). The black arrows show the general directions of the loops from left, then up, down, and finally going to the right. Parameter planes to show the relative area under the curve of GDP (in $1000) when $r = 3$ for different combinations levels of annual health aid, provided from country 2 to country 1, and their timing relative to $A = 0$ in (*c,d*). The x-axis is for the time since the beginning of the epidemic in country 1 (years) and the y-axis is for the annual vaccines received (provided) by country 1 (country 2) in (*c*) (in (*d*)). Figures are shown for country 1 (*a* and *c*) and for country 2 (*b* and *d*). In all of the calculations, epidemic starts in country 1 with 10 initial cases, $k_1(0) = 9$ and $k_2(0) = 55$, and $N_1(0) = N_2(0) = 10^7$. Relative cost of treatment is $\tau = 1000$. The depreciation rates are $\delta_1 = \delta_2 = 0$. Values of the rest of the parameters are given in appendix C and table 1.

There are several limitations of our model, which future studies will consider. For example, in this model, we used one disease to calibrate the model in which case the disease already existed. Also, independent data are required to calibrate some parameters in the model, such as migration and movement of people (e.g. tourism).

Future work will consider the strategic interactions between the two country's health authorities with game theory analysis. Also, our model is scalable, which allows us to consider multi-country scenarios to analyse externality occurring between trading networks or trading blocs. It will more closely emulate real-world situations about health externalities. More importantly, it is recommended in the future to use neoclassical or endogenous growth models to depict the economic growth of the countries whose households are the main decision makers in investment and disease mitigation, see [16,17]. Other extensions of the current modelling study include investment decisions, policy choices and aid allocation.

# 5. Conclusion

Previous research shows strong correlation between perpetual poverty and endemic diseases. Poverty is one of the greatest risk factors for acquiring diseases due to lack of infectious disease controls, infrastructure and sanitation. Infectious disease can affect human capital, resulting in a poverty trap making it difficult for individuals, and by extension entire countries, to escape perpetuating poverty.

This externality contributes to disease persistence and economic decline affecting others beyond the disease outbreak itself.

Simulations from our model of the degree of infectiousness $r$ show two main observations. First is that, *ceteris paribus*, the degree of contagiousness $r$ of diseases affects the economic development in both countries in different ways depending on the demographics and economy strength but also on the friction of moving between countries.

Second, health aid in terms of paid vaccines and logistic support for vaccine campaigns from a rich country to another country with lower socio-economic status can help save both economies from negative externality. Providing health aid, such as vaccination from one country to another altruistically, offers protection to the recipient country with a positive effect on future GDP for both countries. Often governments approach infectious disease management in a siloed manner focusing efforts within their country only—failing to understand the porous nature of their borders to infectious disease transmission through fomites, tourists and domestic travellers entering or leaving the country transiently. Considering externality and incorporating it into health management suggests providing health aid outside the nation state may result in better future economic outcomes through reduced negative externality. Our results highlight the importance of externality in health systems and the significant effect of health aid.

Data accessibility. This article has no additional data.

Authors' contributions. K.J. contributed to model design, analysed the model and wrote the manuscript. T.O. conceived the study, designed and analysed the model and contributed to writing the manuscript. M.G.T. contributed to model analysis, writing and editing of the manuscript. All authors read and approved the final manuscript.

Competing interests. We declare we have no competing interests.

Funding. No funding has been received for this article.

# Appendix A

## A.1. Mathematical model

In the following, we consider two countries sharing a border through which there is influx and outflux of residents who may interact within and between those two countries. We assume that the two populations are otherwise closed.

## A.2. SIR model

For the disease transmission dynamics, we use an SIR model. Let $S_i$, $I_i$ and $R_i$ be the number of susceptible, infectious and recovered people in country $i$, for $i = 1$, 2. Let $N_i = S_i + I_i + R_i$ be the total population size of country $i$.

For each $i = 1$, 2 and $j = 2$, 1,

$$\frac{\mathrm{d}}{\mathrm{d}t}S_i = \mu_i N_i - p_i S_i - S_i\left(\lambda_{i,i}\frac{I_i}{N_i} + \lambda_{i,j}\frac{I_j}{N_j}\right) - \nu_i S_i \tag{A 1}$$

$$\frac{\mathrm{d}}{\mathrm{d}t}I_i = S_i\left(\lambda_{i,i}\frac{I_i}{N_i} + \lambda_{i,j}\frac{I_j}{N_j}\right) - \gamma_i(1 + \Lambda_i)I_i - \nu_i I_i \tag{A 2}$$

and
$$\frac{\mathrm{d}}{\mathrm{d}t}R_i = \gamma_i(1 + \Lambda_i)I_i + p_i S_i - \nu_i R_i, \tag{A 3}$$

with $I_i(0) = I_{i,0}$ and $S_i(0) = N_i - I_i(0)$. And so

$$\frac{\mathrm{d}}{\mathrm{d}t}N_i = \mu_i N_i - \nu_i N_i. \tag{A 4}$$

But since

$$\frac{\mathrm{d}}{\mathrm{d}t}\frac{S}{N} = \frac{S'N - N'S}{N^2} = \frac{S' - (\mu - \nu)S}{N}$$

and similarly

$$\frac{\mathrm{d}}{\mathrm{d}t}\frac{I}{N} = \frac{I'N - N'I}{N^2} = \frac{I' - (\mu - \nu)I}{N},$$

then equations (A 1) and (A 2) become

$$\frac{\mathrm{d}}{\mathrm{d}t}x_i = \mu_i(1 - x_i) - p_i x_i - x_i(\lambda_{i,i}y_i + \lambda_{i,j}y_j) \tag{A5}$$

and

$$\frac{\mathrm{d}}{\mathrm{d}t}y_i = x_i(\lambda_{i,i}y_i + \lambda_{i,j}y_j) - \gamma_i(1 + \Lambda_i)y_i - \mu_i y_i, \tag{A6}$$

where $x = S/N$ and $y = I/N$, while $R/N = 1 - x - y$ makes the third equation in the transformed system of ODE redundant.

## A.3. Solow model

We use Solow model [29] to depict the economic growth in each country $i$ and the spending on health problem denoted by $\varphi$, given as

$$\frac{\mathrm{d}}{\mathrm{d}t}K_i = \hat{\sigma}_{iF}(K_i, L_i) - \delta_i K_i - \varphi(K_i). \tag{A7}$$

Based on the Cobb–Douglas production function $F(K_i, L_i)$ with constant returns to scale [30], the Solow model of the GDP $K_i$ is given by

$$\frac{\mathrm{d}}{\mathrm{d}t}K_i = \hat{\sigma}_i K_i^{\alpha_i} L_i^{1-\alpha_i} - \delta_i K_i - (C_V * (p_i S_i) + C_T * (\gamma_i \Lambda_i I_i)) + \hat{\sigma}_{ij}T(K_i/N_i, N_i; K_j/N_j, N_j)N_j \tag{A8}$$

with $K_i(0) = K_{0,i} > 0$. The number of available labour $L_i = S_i + R_i + \ell(K_i)I_i = N_i - (1 - \ell(K_i))I_i$ is made up from the number of susceptible and recovered individuals in addition to a fraction $\ell(K)$ of infected individuals who would have to work when the GDP is equal to $K$. The model incorporates the following parameters: the elasticity coefficient $\alpha_i > 0$, depreciation rate $\delta_i > 0$, rate of capital accumulation due to production $\hat{\sigma}_i > 0$ and rate of accumulation due to visiting $\hat{\sigma}_{ij} > 0$. The last term in the Solow equation is the revenue due to the human traffic from country $j$. See below for definition of the gravity function $T$.

The *per capita* GDP $k = K/N$ has also

$$\frac{\mathrm{d}}{\mathrm{d}t}\frac{K}{N} = \frac{K'N - N'K}{N^2} = \frac{K' - (\mu - \nu)K}{N}$$

and *per capita* GDP ($1000) is then given by the equation

$$\frac{\mathrm{d}}{\mathrm{d}t}k_i = \sigma_i k_i^{\alpha_i} l_i^{1-\alpha_i} - (\delta_i + \mu_i - \nu_i)k_i - c_V((p_i x_i) + \tau * (\gamma_i \Lambda_i y_i)) + \sigma_{ij}T(k_i, N_i; k_j, N_j)\frac{N_j}{N_i} \tag{A9}$$

with $l = 1 + (\ell(k) - 1)y$, and where $c_V(t) = C_V(t)/1000$, $C_T(t) = \tau * C_V(t)$, $\sigma_i = \hat{\sigma}_i/1000^{1-\alpha_i}$, and $\sigma_{ij} = \hat{\sigma}_{ij}/1000$.

# Appendix B

## B.1. Models parameters

All of the following parameters are functions in the *per capita* GDP ($1000) $k$.

— The birth and death rate are given via

$$\mu N = \mu_0(k) = 36306 - 7207 \log k,$$

with $R^2 = 0.6857$ and

$$\nu N = \nu_0(k) = 10676 - 937.8 \log k,$$

with $R^2 = 0.1091$ and so

$$\mu N - \nu N = 25630 - 6269.2 \log k,$$

and from equation (2.4)

$$N(t) = N(0) + 25630t - 6269.2 \int_0^t \log k(s)\mathrm{d}s,$$

which can be used to find

$$\mu(k) = \frac{36306 - 7207 \log k}{N(0) + 25630t - 6269.2 \int_0^t \log k(s) \mathrm{d}s},$$

$$\nu(k) = \frac{10676 - 937.8 \log k}{N(0) + 25630t - 6269.2 \int_0^t \log k(s) \mathrm{d}s}$$

and

$$\mu(k) - \nu(k) = \frac{25630 - 6269.2 \log k}{N(0) + 25630t - 6269.2 \int_0^t \log k(s) \mathrm{d}s},$$

which is negative for $k > 59.6349$.

Yet it is better to render equation (2.4) as

$$N' = 25630 - 6269.2 \log k(t), \tag{B 1}$$

with the system and use

$$\mu(k(t)) = \frac{36306 - 7207 \log k(t)}{N(t)},$$

$$\nu(k(t)) = \frac{10676 - 937.8 \log k(t)}{N(t)}$$

and

$$\mu(k(t)) - \nu(k(t)) = \frac{25630 - 6269.2 \log k(t)}{N(t)}$$

for simulation and analysis.

— The within country transmission rate is

$$\lambda_{i,i}(k_i) = \lambda_0 \rho(s(k_i)).$$

— The cross border transmission rate is

$$\lambda_{i,j}(k_i, k_j) = \lambda_0 T(k_i, N_i; k_j, N_j) \rho(s(k_i)),$$

where $\lambda_0$ is the basic contact rate and the visitations between country $i$ and $j$, $T(k_i, N_i; k_j, N_j)$, is the spatial interaction modelled using the gravity model of immigration and trade, stemming from Newton's Law of universal gravitation [31],

$$T(k_i, N_i; k_j, N_j) = \frac{N_j^{\xi_0}(k_i/k_j)^{\xi_1}}{N_i^{\xi_2} F_{i,j}^{\xi_3}},$$

where $F_{i,j}$ is the friction (cost, distance, adjacency, common language, colonial link, etc.) between the two countries for people coming from country $j$ into country $i$. (See also [23,32–34].) The elasticity constants of attraction due to GDP and friction are such that $\xi_0, \xi_1, \xi_2, \xi_3 \in \mathbb{R}$.

— The probability of contracting the disease upon contact at sanitation $s$ is

$$\rho(s(k)) = \rho_0(1 - s(k)),$$

and let $\beta_0 = \lambda_0 \rho_0$ be the basic transmission rate, [25].

— The natural recovery rate is

$$\gamma(k) = \gamma(n(k)) = \gamma_0 n(k).$$

See [25].

— The increase in recovery due to treatment is

$$\Lambda(k) = \Lambda_0(1 - \exp(-\phi k)).$$

— The fraction of sick labour available in workforce is $\ell(k)$ where $\ell \in [0, 1]$, with $\ell = 0$ if sick people cannot work.

— The cost of vaccination per person is changing over time due to inflation in prices happening at an annualized inflation rate of $f$ (assumed to be constant; however, see [35]) and is defined as

$$c_V(t) = c_{V,0}(1 + f)^t \quad \text{where } 5 \leq 1000 * c_{V,0} \leq 20.$$

— The cost of treatment per person $c_T(t)$ is a $\tau$-multiple of the cost of vaccination with $100 \leq \tau \leq 2000$.

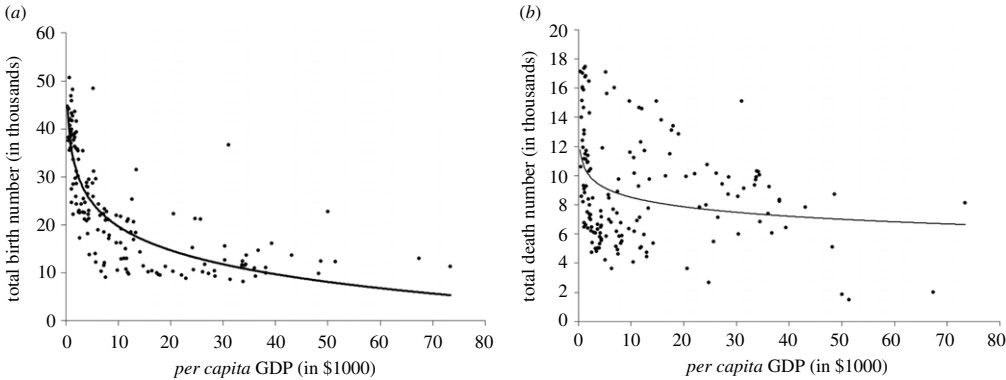

**Figure 9.** (*a*) Total birth number versus *per capita* GDP in $1000. (*b*) Total death number versus *per capita* GDP in $1000.

— The sanitation function is given by[1]

$$\text{logit } s(k) = \theta_{0,s} + \theta_{1,s} \log(k).$$

The parameters are estimated to be $\hat{\theta}_{0,s} = -2.6249$, and $\hat{\theta}_{1,s} = 0.92$, see [25].
— The nutrition function is given by

$$\text{logit } n(k) = \theta_{0,n} + \theta_{1,n} \log(k).$$

The parameters are estimated to be $\hat{\theta}_{0,n} = 1.1771$, and $\hat{\theta}_{1,n} = 0.14$, see [25].
— The probability to voluntarily vaccinate at time $t$ is given by

$$p_i(t) = \theta_p y_i.$$

### B.1.1. External data

The following data were collected from Gapminder.com and analysed using Excel.

A scatter plot for the total birth number against the *per capita* GDP in $1000 for 160 countries for 2008 and the best fit curve are shown in figure 9*a*. It is estimated that

$$\text{total birth number} = -7207 \ln(k/1000) + 36\,306,$$

with coefficient of determination $R^2 = 68.57\%$.

Similarly, a scatter plot for the total death number against the *per capita* GDP in $1000 for 160 countries for 2008 and the best fit curve are shown in figure 9*b*. It is estimated that

$$\text{total death number} = -937.8 \ln(k/1000) + 10\,676,$$

with coefficient of determination $R^2 = 10.91\%$.

# Appendix C

## C.1. Model calibration

Calibration was done via fitting actual data of TB to the model. The goal, though, was not fitting the data for validating the model as a TB, but rather to use some realistic parameter values. To do so, we used the sums of squared differences between the actual and simulated values of the *per capita* GDP ($1000) and the number of infected per 100 000. A genetic algorithm is used to find an optimal solution that minimizes that sums of squares of errors.

Table 1 gives a set of plausible values for the set of parameters of the model.

Figure 10 shows that those values in table 1 can produce patterns similar to real evolution of diseases and economic development in both countries. This does not mean that the changes in both countries' economy are due to the sole externality of TB for both countries.

---

[1]logit $p = \log\,(p/(1-p))$

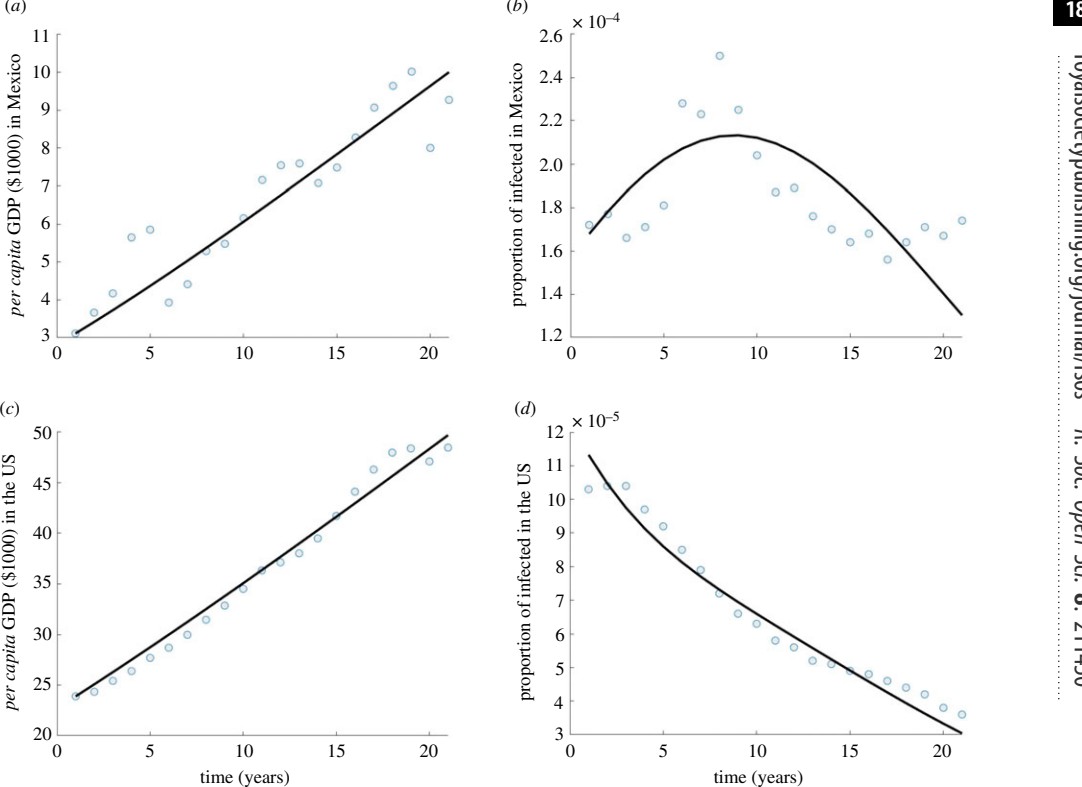

**Figure 10.** Simulation of calibrated models. Simulations of GDP (in $1000) and disease prevalence for country 1 (*a,c*) and country 2 (*b,d*) with values of parameters given in table 1.

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
