## [Peer Review File · Royal Society Open Science]

Review History

RSOS-210732.R0 (Original submission)

Review form: Reviewer 1

Is the manuscript scientifically sound in its present form?

No

Are the interpretations and conclusions justified by the results?

Yes

Is the language acceptable?

Yes

Do you have any ethical concerns with this paper?

No

Have you any concerns about statistical analyses in this paper?

No

Recommendation?

Reject

Comments to the Author(s)

Please see attached review (Appendix A).

Review form: Reviewer 2 (Smit Bhattacharyya)**Is the manuscript scientifically sound in its present form?**

No

Are the interpretations and conclusions justified by the results?

No

Is the language acceptable?

Yes

Do you have any ethical concerns with this paper?

No

Have you any concerns about statistical analyses in this paper?

No

Recommendation?

Major revision is needed (please make suggestions in comments)

Comments to the Author(s)

Attached (see Appendix B).

Decision letter (RSOS-210732.R0)

Dear Dr Jnawali,

The Editors assigned to your paper RSOS-210732 "Mitigation of Externality and Spill-back of Diseases of Poverty through Health Aid" have made a decision based on their reading of the paper and any comments received from reviewers.

Regrettably, in view of the reports received, the manuscript has been rejected in its current form. However, a new manuscript may be submitted which takes into consideration these comments.

We invite you to respond to the comments supplied below and prepare a resubmission of your manuscript. Below the referees' and Editors' comments (where applicable) we provide additional requirements. We provide guidance below to help you prepare your revision.

Please note that resubmitting your manuscript does not guarantee eventual acceptance, and we do not generally allow multiple rounds of revision and resubmission, so we urge you to make every effort to fully address all of the comments at this stage. If deemed necessary by the Editors, your manuscript will be sent back to one or more of the original reviewers for assessment. If the original reviewers are not available, we may invite new reviewers.

Please resubmit your revised manuscript and required files (see below) no later than 29-Nov-2021. Note: the ScholarOne system will 'lock' if resubmission is attempted on or after this deadline. If you do not think you will be able to meet this deadline, please contact the editorial office immediately.

Please note article processing charges apply to papers accepted for publication in Royal Society Open Science (<https://royalsocietypublishing.org/rsos/charges>). Charges will also apply to papers transferred to the journal from other Royal Society Publishing journals, as well as papers submitted as part of our collaboration with the Royal Society of Chemistry (<https://royalsocietypublishing.org/rsos/chemistry>). Fee waivers are available but must be requested when you submit your manuscript (<https://royalsocietypublishing.org/rsos/waivers>).

Thank you for submitting your manuscript to Royal Society Open Science and we look forward to receiving your resubmission. If you have any questions at all, please do not hesitate to get in touch.

on behalf of Dr Derek Abbott (Associate Editor) and Mark Chaplain (Subject Editor)
openscience@royalsociety.org

Reviewer comments to Author:

Reviewer: 1
Comments to the Author(s)
Please see attached review ("RSOS210372.pdf").

Reviewer: 2
Comments to the Author(s)
Attached ("Review report of RSOS.pdf").

===PREPARING YOUR MANUSCRIPT===

Your revised paper should include the changes requested by the referees and Editors of your manuscript. You should provide two versions of this manuscript and both versions must be provided in an editable format:
one version identifying all the changes that have been made (for instance, in coloured highlight, in bold text, or tracked changes);
a 'clean' version of the new manuscript that incorporates the changes made, but does not highlight them. This version will be used for typesetting if your manuscript is accepted.

===PREPARING YOUR REVISION IN SCHOLARONE===

- If you are providing image files for potential cover images, please upload these at this step, and inform the editorial office you have done so. You must hold the copyright to any image provided.
- A copy of your point-by-point response to referees and Editors. This will expedite the preparation of your proof.

- Ensure that your data access statement meets the requirements at <https://royalsociety.org/journals/authors/author-guidelines/#data>. You should ensure that you cite the dataset in your reference list. If you have deposited data etc in the Dryad repository, please include both the 'For publication' link and 'For review' link at this stage.
- If you are requesting an article processing charge waiver, you must select the relevant waiver option (if requesting a discretionary waiver, the form should have been uploaded at Step 3 'File upload' above).
- If you have uploaded ESM files, please ensure you follow the guidance at <https://royalsociety.org/journals/authors/author-guidelines/#supplementary-material> to include a suitable title and informative caption. An example of appropriate titling and captioning may be found at https://figshare.com/articles/Table_S2_from_Is_there_a_trade-off_between_peak_performance_and_performance_breadth_across_temperatures_for_aerobic_scope_in_teleost_fishes_/3843624.

Author's Response to Decision Letter for (RSOS-210732.R0)

See Appendices C & D.

Decision letter (RSOS-211450.R0)

Dear Dr Jnawali,

I am pleased to inform you that your manuscript entitled "Mitigating the Externality of Diseases of Poverty through Health Aid" is now accepted for publication in Royal Society Open Science.

Please ensure that you send to the editorial office an editable version of your accepted manuscript, and individual files for each figure and table included in your manuscript. You can send these in a zip folder if more convenient. Failure to provide these files may delay the processing of your proof.

If you have not already done so, please remember to make any data sets or code libraries 'live' prior to publication, and update any links as needed when you receive a proof to check - for

instance, from a private 'for review' URL to a publicly accessible 'for publication' URL. It is good practice to also add data sets, code and other digital materials to your reference list.

on behalf of Dr Derek Abbott (Associate Editor) and Mark Chaplain (Subject Editor)
openscience@royalsociety.org

Associate Editor Comments to Author (Dr Derek Abbott):
Associate Editor
Comments to the Author:
(There are no comments.)

Reviewer comments to Author:

Appendix A

RSOS-210732

This paper models the interaction between two countries through tourism mobility. An infectious disease with SIR dynamics is prevalent in each country, and the economic structure is modeled through a Solow growth model so that the savings rate is constant. Modeling the role of interaction between two countries taking into account disease prevalence as well as economic activity in understanding pandemics is important.

However, there are some issues in modeling choices that affect what one can learn from the paper about the opening-closing of borders and in health aid that become important in times of pandemics.

On the economics side, the paper uses a Solow model as in Bonds, et al. (2010) while the literature has moved on to using a neoclassical or even endogenous growth models. This is relevant as it is not possible to make welfare comparisons regarding policies in the Solow model.

The epidemiology parameters and vaccination use a rule of thumb when the economic epidemiology literature is studying optimal policies. The optimal policies are non-linear so the use of rule-of-thumb policies which depend on the capital stock is misleading and does not capture the complexity of how policies change with incidence of the disease and economic conditions.

There is lack of clarity on how disease externality is modeled and dealt with. There is a different disease externality in Gersovitz and Hammer (2003) and Goenka and Liu (2020) but the externality in this paper, as I understand it, is the movement of people across countries. It is not at all clear how an ad-hoc aid can correct it as it does not correct for the marginal distortion caused by mobility.

The main issue is that there is no choice in the model – how investment is done, how policy choices are made, how visitation depends on disease incidence, how vaccination is done, how aid is allocated.

On the epidemiology side it is not clear how the movement of infective and susceptible is being modeled. It should take into account the movement of new susceptibles as well as recovered and recovered across countries. The foreign recovered and vaccinated visitors will change the size of the population in a country so that the fraction of susceptibles (now includes both domestic population as well as visitors) and infectives (same applies) will change. Thus, I think that the equations for the population sizes are missing and the rate of visitation should also be there in the epidemiology equations in the model.

Appendix B

Review report of RSOS-210732

In this paper, authors have considered the spill back of disease of poverty using two country model and discussed how health aid can be helpful to mitigate the externality that comes from the disease. Authors have developed the model, calibrated parameters and performed numerical analysis to illustrate the qualitative aspects of how health aid be helpful in reducing the disease externality. The paper certainly considers an important issue in disease control as well as economic downfall, but it needs some more rigorous work before it should go to publication house.

The major points are following:

1. First of all, I'm little skeptical about the word used in title. the authors have used the word 'spillback'. The **'spillback' has some technical meaning in the theory of Epidemiology**. However, it has not been focused or discussed in the context of spillback of the disease in the rest of the paper. The model also does not describe the dynamic 'spillback process'. It gave me a wrong impression and will do for future readers as well. Author may replace it by simpler word like 'cross-border infection' etc.
2. For example, can authors estimate or calculate **what is the potential spillback risk because of the cross-border infection and how much health aid is required to save the country from economic downfall?** What I see presently in the paper is just a qualitative discussion of what affects what.
3. I have also doubt about one conclusion that author made in the paper. **They say that high friction affects the GDP of the lower income country more than the higher income country. But I think it depends on that disease intensity in the two countries**. For example, here authors do not compute any epidemiological indicator to measure the disease intensity of the two countries like R_0 , the basic reproduction ratio. So, if one considers this basic reproduction ratio different for the two different countries in the prescribed range for this particular infection, the conclusion may be different. Authors definitely needs to consider in their analysis.

4. **The 'introduction' does not seem to be very motivated according to the results that author wanted to discuss in their paper.** So the 'introduction' should be rewritten.
5. In the introduction authors have mentioned diseases like TB, malaria or HIV. **I believe the model formulated in the paper is way more simple to describe any of such infections.** So, authors should highlight what specific infection they're talking about, which also relates the point 6 below
6. Another major drawback in this paper I see **the disparity in timescale of disease and timescale of economic downfall as it shows in figure 1 and in many others.** The duration of an outbreak is 20 years minimum in some figures, and it is more than 70 or 80 years. I'm not sure what kind of disease or infection that author had in their mind. This is a major drawback and authors need to rescale in their code and redo all simulations.
7. Some figures have multiple legends, but not reflected in the figure template.
8. Another important issue is rewriting the calibration process of parameters of the model (in Appendix). Authors have used several functional forms to calibrate their parameters. Although some references have been pointed out, but **it requires more details for naive readers.**

I would be happy to see the revision taking care of all these points.

Appendix C

RSOS-210732

This paper models the interaction between two countries through tourism mobility. An infectious disease with SIR dynamics is prevalent in each country, and the economic structure is modeled through a Solow growth model so that the savings rate is constant. Modeling the role of interaction between two countries taking into account disease prevalence as well as economic activity in understanding pandemics is important.

Answer: We would like to thank the reviewer for finding the research problem important. Many of the comments and suggested changes from reviewer 1 envision a much bigger, differently focused, and much more expansive economic policy paper. It is difficult to address such comments when the intention of our first paper was to introduce our model and apply it to a health and health policy (not economic) externality context. Our future work will consider many of the ideas and have more emphasis on economic policy considerations.

However, there are some issues in modeling choices that affect what one can learn from the paper about the opening-closing of borders and in health aid that become important in times of pandemics.

On the economics side, the paper uses a Solow model as in Bonds, et al. (2010) while the literature has moved on to using a neoclassical or even endogenous growth models. This is relevant as it is not possible to make welfare comparisons regarding policies in the Solow model.

Answer: We are aware that the current model is a crude depiction of welfare and obtaining accurate welfare comparisons for policy making was not the focus of the modeling exercise. Our model doesn't extend to welfare comparisons while it is a direction in economical modeling of the effect of diseases, it would require adding an optimization step and consumption data and that will require a large change in the work that started as an extension to the work of Bonds et al. (2010). We also added a sentence to the discussion section to indicate the future research direction of including welfare comparisons along the lines of Gersovitz and Hammer (2003) and Goenka and Liu (2020). The additional text is:

“Future work will consider the strategic interactions between the two country's health authorities with game theory analysis. Also, our model is scalable which allows us to consider multi-country scenarios to analyze externality occurring between trading networks or trading blocs. It will more closely emulate real world situations about health externalities. More importantly, it is recommended in the future to use neoclassical or endogenous growth models to depict the economic growth of the countries whose households are the main decision makers in investment and disease mitigation, see [12, 14]. Other extensions of the current modeling study include investment decisions, policy choices, and aid allocation”.

We have also included references: Gersovitz and Hammer (2003) and Goenka and Liu (2020) to the introduction as work on the issue for single countries.

The epidemiology parameters and vaccination use a rule of thumb when the economic epidemiology literature is studying optimal policies. The optimal policies are non-linear so the use of rule-of-thumb policies which depend on the capital stock is misleading and does not capture the complexity of how policies change with incidence of the disease and economic conditions.

Answer: We made assumptions regarding epidemiology and vaccination parameters based on *voluntarily* vaccination as many diseases like TB in many places, and we did not consider the additional policy implications (e.g. vaccination uptake and hesitancy, etc.). These are great ideas to consider but

they are outside the scope of the paper which is intended to be a simple modeling exercise to show the effect of disease externality on the economy and how aids (in the form of vaccine subsidy) can result a change in the effect of the externality. Of course, those are smart new extensions that we can add to our current model. We will consider such extensions in a follow-on manuscript and as part of our future research in to this topic. We added the idea that vaccinations are *voluntarily* and *non-mandatory* in many places of the text.

There is lack of clarity on how disease externality is modeled and dealt with. There is a different disease externality in Gersovitz and Hammer (2003) and Goenka and Liu (2020) but the externality in this paper, as I understand it, is the movement of people across countries. It is not at all clear how an ad-hoc aid can correct it as it does not correct for the marginal distortion caused by mobility.

Answer: In our paper, we follow the definition of externality in its economic sense (from Alvarez et al., 2020) that is consistent with the following definition,

"Externalities are net costs (negative externalities) or benefits (positive externalities) that a person's behavior imposes on other people for which he does not account when deciding how to behave. In the context of infectious disease, behaviors that may create externalities are those that affect other people's risk of infection (Alvarez et. al, 2020). "

Alvarez, F. E., Argente, D. & Lippi, F. (2020) A simple planning problem for COVID-19 lockdown. NBER Working Paper No. 26981.

So externality is the economic loss as well as infections incurred by country 2 due to the spread of disease in country 1 because of country 1 's incapability of protect its people through economic prosperity that gives rise to better hygiene, nutrition and treatment (Figure 1 e and f shows a proof). Mobility constitutes the medium of the spread of the disease, and air in many environmental externality problems, it cannot not be always decreased through higher friction (Figure c and d as a proof). It could be however abated through health aid in the form of vaccination.

To better clarify this point we also explained the concept of externality in view of the aforementioned paper (and also added it as a reference) with the following sentence that is added to the introduction:

"The term externality is defined in a number of ways. Externality is a result when an agent produces or consumes a good or service that causes a cost or benefit on other parties who are not involved in this transaction [7]. *In terms of disease spread, it could be identified through people's risk of infection due to other people's behaviors [1].*" (Alvarez et al., 2020 reference).

The main issue is that there is no choice in the model – how investment is done, how policy choices are made, how visitation depends on disease incidence, how vaccination is done, how aid is allocated.

Answer: We thank the reviewer for proposing those interesting extensions to the current modeling approach. This is an initial paper that focuses on the effect of health aid to remedy externality in contrast to closing borders, especially if there are mutual benefits. We added a sentence in the discussion section to express the need to expand the model to address those points for future research efforts. We are also considering developing a follow up paper that will address these points (investment, policy choices, etc.).

The sentence is:

Future work will consider. ... Other extensions of the current modeling study include investment decisions, policy choices, and aid allocation.

On the epidemiology side it is not clear how the movement of infective and susceptible is being modeled. It should take into account the movement of new susceptibles as well as recovered and recovered across countries. The foreign recovered and vaccinated visitors will change the size of the population in a country so that the fraction of susceptibles (now includes both domestic population as well as visitors) and infectives (same applies) will change. Thus, I think that the equations for the population sizes are missing and the rate of visitation should also be there in the epidemiology equations in the model.

Answer: The reviewer makes a good point. The interaction between the two countries' citizens is modeled using the Newton's law of attraction. The number of visitors to the total population cannot be of significant alteration to the total population since visitations happen both ways and are transient in nature. Also, there are high (peak tourism) and low (off peak) seasons during the year and so we take the assumption that, on average over the year, there will be no significant change in the total population made by visitation of one single country to another. Also, it will be distributed over the whole country and we doubt that it will be of practical influence. Note that visitation here is reciprocal and is accounted for one single country in our model.

Appendix D

Review report of RSOS-210732

In this paper, authors have considered the spill back of disease of poverty using two country model and discussed how health aid can be helpful to mitigate the externality that comes from the disease. Authors have developed the model, calibrated parameters and performed numerical analysis to illustrate the qualitative aspects of how health aid be helpful in reducing the disease externality. The paper certainly considers an important issue in disease control as well as economic downfall, but it needs some more rigorous work before it should go to publication house.

The major points are following:

1. First of all, I'm little skeptical about the word used in title. the authors have used the word 'spillback'. The **'spillback' has some technical meaning in the theory of Epidemiology**. However, it has not been focused or discussed in the context of spillback of the disease in the rest of the paper. The model also does not describe the dynamic 'spillback process'. It gave me a wrong impression and will do for future readers as well. Author may replace it by simpler word like 'cross-border infection' etc.

Answer: We agree with the reviewer, and we have edited the title to remove the term "spillback." We changed it to "cross-border infection."

2. For example, can authors estimate or calculate **what is the potential spillback risk because of the cross-border infection and how much health aid is required to save the country from economic downfall?**

What I see presently in the paper is just a qualitative discussion of what affects what.

Answer: We added some quantitative results in figures 7 and 8 besides those in figure 6 and including a new paragraph in between those figures. They are bolstering the main finding of the paper that swift and higher levels of aid can save both countries almost as much as when decreasing risk of infection, by for example social distances that are not reasonable over decades.

3. I have also doubt about one conclusion that author made in the paper. **They say that high friction affects the GDP of the lower income country more than the higher income country. But I think it depends on that disease intensity in the two countries.** For example, here authors do not compute any epidemiological indicator to measure the disease intensity of the two countries like R_0 , the basic reproduction ratio. So, if one

considers this basic reproduction ratio different for the two different countries in the prescribed range for this particular infection, the conclusion may be different. Authors definitely needs to consider in their analysis.

Answer: It is part of each. But first, in this model, rates are time-dependent and it is not advisable to calculate the basic reproduction number R_0 to indicate a threshold for disease free equilibrium. But we know that the basic reproduction number R_0 is directly proportional to the transmission rate. That means that figure 3 could confirm our conclusion. In Figure 3 (a) low transmission rate was worse than higher ones at low friction and it had relatively stronger effect on country 1's economy than on country 2's economy, Figure 3 (b). But, in Figure 3 (c) and (d), with higher friction, the drop of country 1's economy is apparently worse than that of country 2's economy. That whole effect is corrected with large aid. One possible explanation of that is, the higher friction doesn't affect the higher income country since they have better sanitation and hygiene (infrastructure also) to deal with the disease and economy loss from border closure and for example the attraction model is not reciprocal for migrant labor as more labor from the low-income country will try to go to the high income country and not vice-versa.

4. The 'introduction' does not seem to be very motivated according to the results that author wanted to discuss in their paper. So the 'introduction' should be rewritten.

Answer: We reviewed and compared our Introduction section to the Discussion and Conclusion sections to address Reviewer 2's comment regarding a potential mis-match of content. We do not find the topics discussed in the introduction to be mis-matched to the interpretation of the results in the Discussion section or the Conclusion section.

Table: Summary of Issues discussed in the manuscript.

Introduction	Discussion	Conclusion
 Describe diseases of poverty 	 Context of externality for health 	 Poverty and correlation to endemic diseases

 • Labor participation and infectious disease 	 • Description to country disease economy model 	 • Degree of infectiousness r and observations (Ceteris paribus and health aid effects)
 • HIC and LIC infrastructure for surveillance 	 • Baseline scenario with parameter variation 	 • Government and porous borders
 • Definition poverty trap in infectious disease context 	 • Variations from the baseline scenario 	 • Incorporating externality into health management
 • Definition of externality 	 • Limitations of the model 	 • Importance of externality and effect of health aid
 • Use of modelling for health externality 	 • Future work 	
 • Description of disease transition model 		

Regardless of the lack of mismatch we interpret the reviewer’s comment to mean that there is information regarding vaccines and health aid (the main bulk of the Conclusion section) that does not appear to be addressed in the introduction. We have added new text to the introduction to cover these missing ideas to better reflect the ideas presented in the conclusion and provide a better match. We thank the reviewer for the comment that has helped us to improve the continuity of our manuscript.

New introduction text (inserted after reference [4]):

“It is well known that poor people in low-income countries suffer from higher rates of illness particularly infectious diseases and malnutrition due to several factors including little food, unclean water, low levels of sanitation and shelter, lack of infrastructure to deal with high exposure to infectious agents and lack of appropriate medical care [5]. Research has confirmed that addressing determinants of health can yield significant sustainable returns for improving population health [6]. The ability of the health care system to provide effective services can be strengthened by various types of health aid such as training (increasing the availability of adequate skilled health care workers e.g., nurses, doctors), infrastructure (essential equipment, supplies, health facilities) and medicines (such as vaccines, antivirals, and drugs) to meet the

needs of the population they assist [5]. Our analysis focuses on the provision of health aid given in the form of vaccines as an example. This health aid addresses and manages longer-term transmissible disease impacts in the context of our model where externality is caused by friction (the movement of infected people) between two countries."

New citations were added to the References of the manuscript:

[5] Christopher Garimoi Orach. Health equity: challenges in low income countries. *African health sciences*, 9 Suppl 2(Suppl 2), S49–S51, 2009.

[6] Alan D Lopez, Colin D Mathers, Majid Ezzati, Dean T Jamison, Christopher J L Murray. Global burden of disease and risk factors. New York: The World Bank and Oxford University Press, 2006.

5. In the introduction authors have mentioned diseases like TB, malaria or HIV. **I believe the model formulated in the paper is way more simple to describe any of such infections.** So, authors should highlight what specific infection they're talking about, which also relates the point 6 below.

Answer: please see our answer to number 6.

6. Another major drawback in this paper I see **the disparity in timescale of disease and timescale of economic downfall as it shows in figure 1 and in many others.** The duration of an outbreak is 20 years minimum in some figures, and it is more than 70 or 80 years. I'm not sure what kind of disease or infection that author had in their mind. This is a major drawback and authors need to rescale in their code and redo all simulations.

Answer: We do not require recoding, new simulations, or redoing analysis. The infection enters country 1 and becomes endemic. We have added the following text and reference to the introduction of the paper to better explain the simulations and infections and how they relate to the x-axis (time in years) on most of the figures. The text is: "Under our simulations, a non-specific infection is shown entering the country where it becomes endemic circulating within the population repeatedly. For example, a non-specific infection could be a disease of poverty such as tuberculosis (TB), which has been circulating in India for at least a few hundred years (Sandhu, 2011)."

Sandhu G. K. (2011). Tuberculosis: current situation, challenges and overview of its control programs in India. *Journal of global infectious diseases*, 3(2), 143–150. <https://doi.org/10.4103/0974-777X.81691>

7. Some figures have multiple legends, but not reflected in the figure template.

Answer: We fixed the legends of all of the figures to reflect the shown curves (both red and blue).

8. Another important issue is rewriting the calibration process of parameters of the model (in Appendix). Authors have used several functional forms to calibrate their parameters. Although some references have been pointed out, but **it requires more details for naive readers.**

Answer: We added some more details to the calibration process in Appendix III as requested.

I would be happy to see the revision taking care of all these points.